# *Rhodobacter capsulatus* forms a compact crescent-shaped LH1–RC photocomplex

Kazutoshi Tani [1,9] ✉, Ryo Kanno[2,3,9], Xuan-Cheng Ji[4], Itsusei Satoh[4], Yuki Kobayashi[4], Malgorzata Hall[2], Long-Jiang Yu [5], Yukihiro Kimura [6], Akira Mizoguchi[1], Bruno M. Humbel[2,7], Michael T. Madigan[8] & Zheng-Yu Wang-Otomo [4] ✉

*Rhodobacter* (*Rba.*) *capsulatus* has been a favored model for studies of all aspects of bacterial photosynthesis. This purple phototroph contains PufX, a polypeptide crucial for dimerization of the light-harvesting 1–reaction center (LH1–RC) complex, but lacks protein-U, a U-shaped polypeptide in the LH1–RC of its close relative *Rba. sphaeroides*. Here we present a cryo-EM structure of the *Rba. capsulatus* LH1–RC purified by DEAE chromatography. The crescent-shaped LH1–RC exhibits a compact structure containing only 10 LH1 αβ-subunits. Four αβ-subunits corresponding to those adjacent to protein-U in *Rba. sphaeroides* were absent. PufX in *Rba. capsulatus* exhibits a unique conformation in its N-terminus that self-associates with amino acids in its own transmembrane domain and interacts with nearby polypeptides, preventing it from interacting with proteins in other complexes and forming dimeric structures. These features are discussed in relation to the minimal requirements for the formation of LH1–RC monomers and dimers, the spectroscopic behavior of both the LH1 and RC, and the bioenergetics of energy transfer from LH1 to the RC.

The genus *Rhodobacter* (*Rba.*) is composed of a highly heterogeneous group of anoxygenic purple nonsulfur phototrophs with 16 validly named species[1]. Among these, *Rba. capsulatus* along with *Rba. sphaeroides* have been widely used as models for both fundamental and applied studies in photochemistry, genetics, metabolism, and regulation. *Rba. capsulatus* in particular has been a workhorse because of its extensive metabolic diversity and facile genetics[2]. According to a recent proposal[1,3], the genus *Rhodobacter* can be grouped into four major monophyletic clusters based on taxogenomic analyses. *Rba. capsulatus* falls into Clade II, species of which lack cardiolipin (CL) and glycolipids[1]. By contrast, *Rba. sphaeroides* falls into Clade I, species of

which contain CL and glycolipids and contain larger genomes with higher G + C contents than species in Clade II.

The photosynthetic gene cluster in *Rhodobacter* species contains a unique gene, *pufX*, which encodes the polypeptide PufX present in the light-harvesting 1–reaction center (LH1–RC) core complex. PufX in *Rba. sphaeroides* has been demonstrated to play a critical role in dimerization of the LH1–RC complex[4–9]. By contrast, *Rba. capsulatus* forms only a monomeric LH1–RC despite the presence of PufX[10], and a similar result has been observed for *Rba. veldkampii*[11–14]. In addition, the *Rba. sphaeroides* LH1–RC contains a newly identified protein named protein-U[8,15] (equivalent to protein-Y[7,16] or PufY[9]), while both

[1]Graduate School of Medicine, Mie University, Tsu, Japan. [2]Scientific Imaging Section, Research Support Division, Okinawa Institute of Science and Technology Graduate University (OIST), 1919-1Tancha, Onna-Son, Kunigami-Gun, Okinawa, Japan. [3]Quantum wave microscopy unit, Okinawa Institute of Science and Technology Graduate University (OIST), 1919-1Tancha, Onna-Son, Kunigami-Gun, Okinawa, Japan. [4]Faculty of Science, Ibaraki University, Mito, Japan. [5]Photosynthesis Research Center, Key Laboratory of Photobiology, Institute of Botany, Chinese Academy of Sciences, Beijing, China. [6]Department of Agrobioscience, Graduate School of Agriculture, Kobe University, Nada, Kobe, Japan. [7]Department of Cell Biology and Neuroscience, Juntendo University, Graduate School of Medicine, Tokyo, Japan. [8]School of Biological Sciences, Department of Microbiology, Southern Illinois University, Carbondale, IL, USA. [9]These authors contributed equally: Kazutoshi Tani, Ryo Kanno ✉e-mail: ktani@doc.medic.mie-u.ac.jp; wang@ml.ibaraki.ac.jp

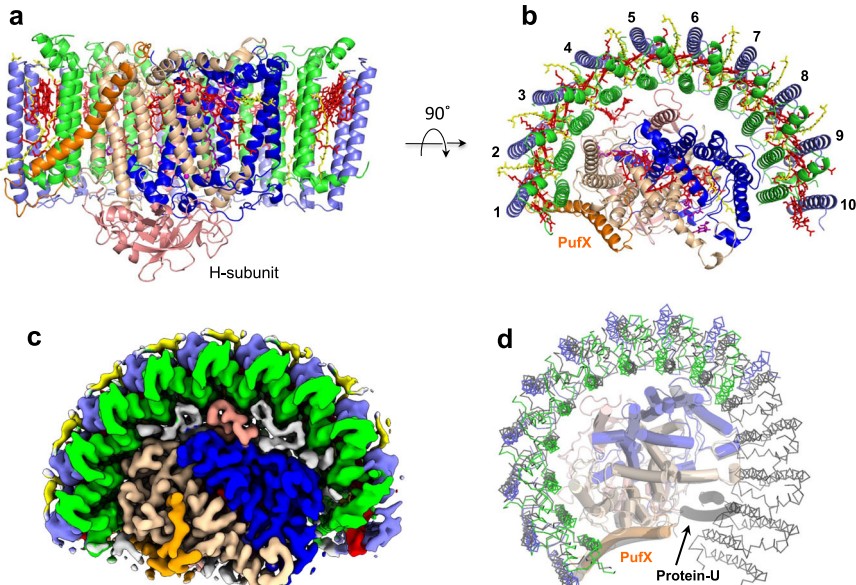

**Fig. 1 | Structure overview of the *Rba. capsulatus* LH1–RC. a** Side view of the core complex parallel to the membrane plane. Color scheme: LH1-α, green; LH1-β, slate blue; PufX, orange; RC-L, wheat; RC-M, blue; RC-H, salmon; BChl *a*, red sticks; spheroidenes, yellow sticks; BPhe *a*, magenta sticks. **b** Top view from periplasmic side of the membrane with the same color scheme as in (**a**). **c** Top view of surface representation for the LH1–RC from the periplasmic side. Lipids and detergents are colored in gray. **d** Overlap view of the *Rba. capsulatus* LH1–RC (colored) and that of *Rba. sphaeroides* (black, PDB: 7F0L) by superposition of Cα carbons of the RC-M subunits. Protein-U as it exists in the *Rba. sphaeroides* LH1–RC is shown by a black transparent cartoon.

*Rba. capsulatus* and *Rba. veldkampii* lack this protein[15]. The absence of protein-U in a protein-U mutant derivative of *Rba. sphaeroides* resulted in a reduced amount of the dimeric complex, and the monomeric LH1–RC from this mutant revealed a crescent-shaped structure containing 10–13 αβ-polypeptides (subunits)[8,9]; this compares with the 14 subunits present in the *Rba. sphaeroides* wild-type monomeric complex[9,15,16]. Collectively, these data pointed to a crucial role for protein-U in controlling the number of αβ-subunits required for correct assembly and stabilization of the LH1–RC dimer. This conclusion is also supported by a structural analysis of the LH1 from *Rba. veldkampii*, a protein-U-lacking complex that consists of 15 αβ-subunits and which forms a C-shaped monomeric LH1–RC structure[14].

To further investigate structural and functional roles of protein-U and PufX, we have determined the structure of the native *Rba. capsulatus* LH1–RC complex. Thus far, no atomic structure is known for either the LH1 or RC from *Rba. capsulatus*, despite the latter being a well-studied model of purple bacterial photochemistry and spectroscopy[17,18]. Surprisingly, and differing from all wild-type LH1s with known structures (obtained from ~10 species), we have found that the current *Rba. capsulatus* LH1 structure in an RC-associated form contains only 10 αβ-subunits, making it the smallest LH1 complex reported by far. Our results thus reveal that wild-type purple bacteria can produce photosynthetically competent LH1 complexes containing significantly fewer than 14 αβ-subunits and highlights the structural diversity of purple bacterial LH1–RCs.

## Results
### Structural overview
The LH1–RC purified from *Rba. capsulatus* wild-type strain NBRC16435[T] using two-step solubilization followed by DEAE chromatography exhibited an absorption maximum ($Q_y$) at 882 nm for LH1 (Supplementary Fig. 1), and the cryo-EM structure of the core complex was determined at 2.62 Å resolution (Fig. 1, Supplementary Table 1 and Supplementary Figs. 2–5). The *Rba. capsulatus* monomeric LH1–RC forms a crescent-shaped structure with LH1 containing 10 αβ-subunits distributed on one side of the RC with the other side of the RC completely exposed to the membrane region (Fig. 1b, c). Focused 3-D

classification[19,20] was conducted iteratively on a localized region containing the gap of the LH1 ring and three αβ-subunits at one end of the LH1 crescent until convergence. Four classes of the 3-D maps were obtained, each of which showed similar conformations independent of resolution (Supplementary Fig. 4). By checking these maps, only a single class in each form showed well-resolved density corresponding to the three LH1 subunits, while other maps were relatively featureless yet sufficiently resolved to show that an extra LH1 subunit was not accommodated.

It is notable that the number of LH1 αβ-subunits is the fewest by far of any core light-harvesting complex from an anoxygenic phototroph. As a result of this small number of LH1 subunits, interactions between LH1 and the RC in *Rba. capsulatus* are less extensive than those in LH1–RCs composed of a closed or small-gapped LH1 ring where up to 17 LH1 subunits may be present. This was evident from observations that unlike in *Rba. sphaeroides*, an LH1-only complex was easily isolated from *Rba. capsulatus* membranes during solubilization and purification (Supplementary Fig. 1a) and that ~5 % of the isolated complexes were LH1-only (Supplementary Fig. 3a). PufX is located at one edge of the *Rba. capsulatus* LH1 crescent and interacts with one pair of αβ-polypeptides and the RC-L subunit (Fig. 1b).

Comparison of the *Rba. capsulatus* LH1–RC with that of *Rba. sphaeroides* revealed significant differences. The four LH1 αβ-subunits near protein–U in the *Rba. sphaeroides* monomeric complex are absent (along with protein-U itself) from the *Rba. capsulatus* LH1 (Fig. 1d); this supports the proposed role of protein–U in controlling the number of LH1 αβ-subunits in the *Rba. sphaeroides* complex[8,9]. Many residual densities were observed in the cryo-EM density map of the *Rba. capsulatus* LH1–RC and assigned to phospholipids and detergents (Supplementary Fig. 6).

### Cofactors in the *Rba. capsulatus* LH1–RC
Twenty-five bacteriochlorophylls (BChl) *a*, eighteen spheroidenes, two bacteriopheophytins, four ubiquinones (UQ-10) and one nonheme Fe were identified in the *Rba. capsulatus* LH1–RC (Fig. 2). In addition to the BChls *a* assigned to either LH1 or RC, an isolated BChl *a* was identified from our cryo-EM density map in the inner space between

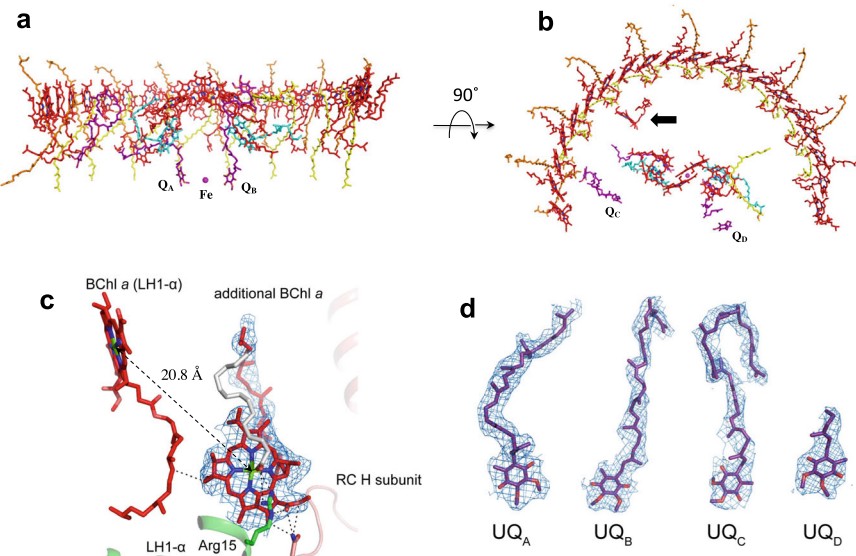

**Fig. 2 | Arrangement of the cofactors in the *Rba. capsulatus* LH1–RC. a** Side view along the membrane plane with the periplasm above and the cytoplasm below. Color scheme: BChl *a*, red sticks; spheroidenes (group-A), yellow sticks; spheroidenes (group-B), orange sticks; BPhe *a*, magenta sticks; UQ-10, purple sticks; non-heme Fe, magenta sphere. **b** Top view from periplasmic side of the membrane with the same color scheme as in (**a**). One additional BChl *a* and two additional ubiquinones ($Q_C$ and $Q_D$) were detected in the LH1–RC structure. Arow indicates the extra BChl *a*. **c** Close contacts (<4.0 Å) between the extra BChl *a* and surrounding residues, and Mg–Mg distance between the extra BChl *a* and the nearest LH1-BChl *a*. Color scheme as in Fig. 1a except for myristate in gray. **d** All four of ubiquinones ($Q_A$, $Q_B$, $Q_C$ and $Q_D$) were detected in the LH1–RC structure. The density maps are shown at a contour level of 5.0σ.

LH1 and RC on the cytoplasmic side (Fig. 2b). This BChl *a* is surrounded by two LH1 α-polypeptides, the RC H-subunit, a phospholipid and a detergent molecule, and is coordinated from the back by a putative myristate at its central Mg (Fig. 2c, Supplementary Fig. 5b). This "extra" BChl *a* interacts with nearby Asn52 of the RC H-subunit and Arg15 of an LH1 α-polypeptide through its C13[1] and C13[3] carbonyl groups, respectively, by forming stable hydrogen bonds (Fig. 2c). Interestingly, the *Rba. veldkampii* LH1–RC contains an extra bacteriopheophytin (BPhe) *a* molecule[14], but its location differs from that of the extra BChl *a* in the *Rba. capsulatus* complex (Supplementary Fig. 7). Although the function of this extra BChl *a* in the *Rba. capsulatus* LH1–RC is unknown, it may play a role in light-harvesting and/or excitation energy transfer considering its nearest distances from an LH1 BChl *a* (20.8 Å Mg–Mg) and from the accessary BChl *a* (28.6 Å Mg–Mg) and the BPhe *a* (21.5 Å edge-to-edge of the macrocycles) in the RC A-branch.

As for the *Rba. sphaeroides* LH1[9,15,16], two groups of carotenoids were identified in the *Rba. capsulatus* LH1 complex (Fig. 2a, b) and confirmed from their absorption spectra (Supplementary Fig. 1a). Nine all-*trans* spheroidenes (group-A) are deeply embedded in the transmembrane region between αβ-subunits while eight all-*trans* spheroidenes (group-B) protrude on the periplasmic surface. In addition to the conserved UQ-10 at the $Q_A$ and $Q_B$ sites, two ubiquinones were detected in the *Rba. capsulatus* LH1–RC structure (Fig. 2b, d): one ($Q_C$) was located close to PufX with its head group pointing to the cytoplasmic side while the second ($Q_D$) was present on the membrane-exposed side of the RC with its head group near the periplasmic surface.

## PufX in the *Rba. capsulatus* LH1–RC

The full-length PufX of *Rba. capsulatus* contains 78 amino acids (Supplementary Fig. 8) in which the N-terminal Met and C-terminal 12 residues were invisible in our cryo-EM density map, presumably due to either post-translational modification or disordered conformation. The expressed *Rba. capsulatus* PufX is reported to contain 68 residues of molecular mass of 7490 Da[21], however such was not well resolved in our mass spectroscopic analyses (Supplementary Fig. 8c).

Although the transmembrane domain of *Rba. capsulatus* PufX shows a helical structure resembling that of PufX from other *Rhodobacter* species (Fig. 3a), significantly different conformations were found for the N-terminal region (Fig. 3b), which is known to play a crucial role in dimerization of the *Rba. sphaeroides* LH1–RC[6,8,22]. PufX in the *Rba. capsulatus* LH1–RC has a "turn-up" conformation in its N-terminal region (Ser2–Asn12) that self-associates with its own transmembrane domain (Trp21–Tyr28) and interacts with nearby LH1 α- and β-polypeptides (Fig. 3c, d). Sequence comparisons of *Rba. capsulatus* PufX with that of *Rba. sphaeroides*, a species that forms both dimeric and monomeric LH1–RCs[7–9,15,16], revealed low similarities (Supplementary Fig. 8a) in the N-terminal region, and this was also true for *Rba. veldkampii* PufX, which contains a relatively short N-terminus and from which only monomeric LH1–RC has been isolated[12,14]. These findings highlight the broad structural diversity of PufX, even within a single genus of purple bacteria, and support the conclusion that the N-terminus of PufX is responsible for forming the dimeric core complex of *Rba. sphaeroides*[4–9].

## The *Rba. capsulatus* RC

As expected from sequence comparisons (Supplementary Fig. 9), overall structures of the *Rba. capsulatus* RC proteins were quite similar to those from *Rba. sphaeroides*[9,15,16,23,24], with a root-mean-square deviation (RMSD) of 0.42 Å for the mainchain Cα carbons (PDB: 7F0L) (Fig. 4a). Major cofactors in the *Rba. capsulatus* RC were generally superimposable with those of the *Rba. sphaeroides* RC (Fig. 4b) but exhibited local differences. For example, the distance between the *Rba. capsulatus* special pair BChls *a* (P) is 7.65 Å (Mg–Mg distance), shorter than that in *Rba. sphaeroides* (7.85 Å) (Supplementary Table 2), whereas the coordination lengths between the ligating His residues and special pair BChls *a* are longer than those in *Rba. sphaeroides* (Supplementary Table 2). It is also notable that the amino acid adjacent to the special pair BChl *a*-coordinating His (L174) in the *Rba. capsulatus* RC-L subunit is a Phe (L173) instead of the Ala present in the *Rba. sphaeroides* protein (Fig. 4c). The Phe in *Rba. capsulatus* lies close (~3.7 Å) to a nearby Trp (L244), enabling π–π interactions between these two aromatic residues (Fig. 4c).

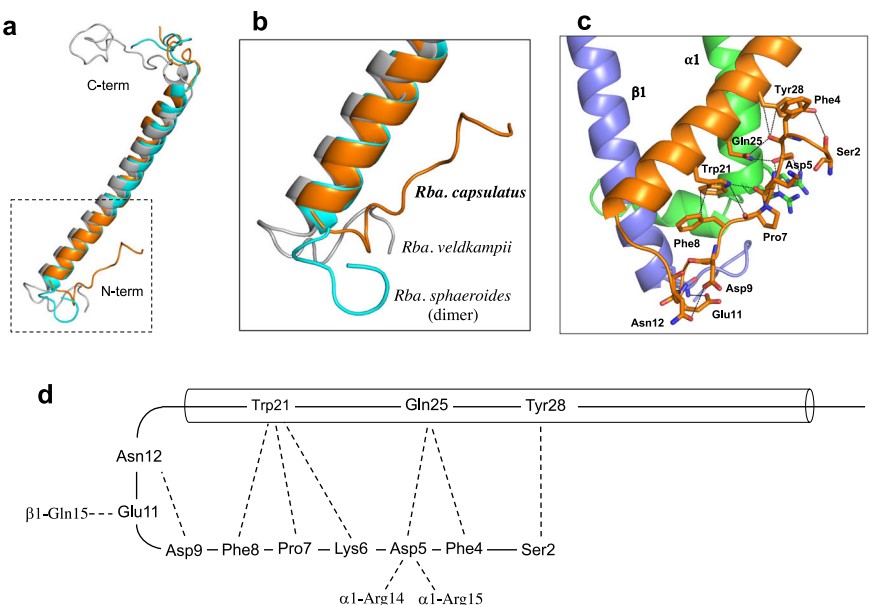

**Fig. 3 | PufX in the *Rba. capsulatus* LH1–RC. a** Superpositions of the *Rba. cap-sulatus* PufX (orange) with those from *Rba. sphaeroides* (cyan, PDB:7VY2) and *Rba. veldkampii* (gray, PDB: 7DDQ). **b** Expanded view of the N-terminal regions of PufX marked by the dashed box in (**a**). **c** Interactions (<4.0 Å) of the PufX N-terminus (orange) with the amino acids in its own transmembrane domain and nearby LH1 α(green)- and β(slate blue)-polypeptides. **d** Schematic representation of self-association of the PufX N-terminus and interactions with surrounding polypeptides.

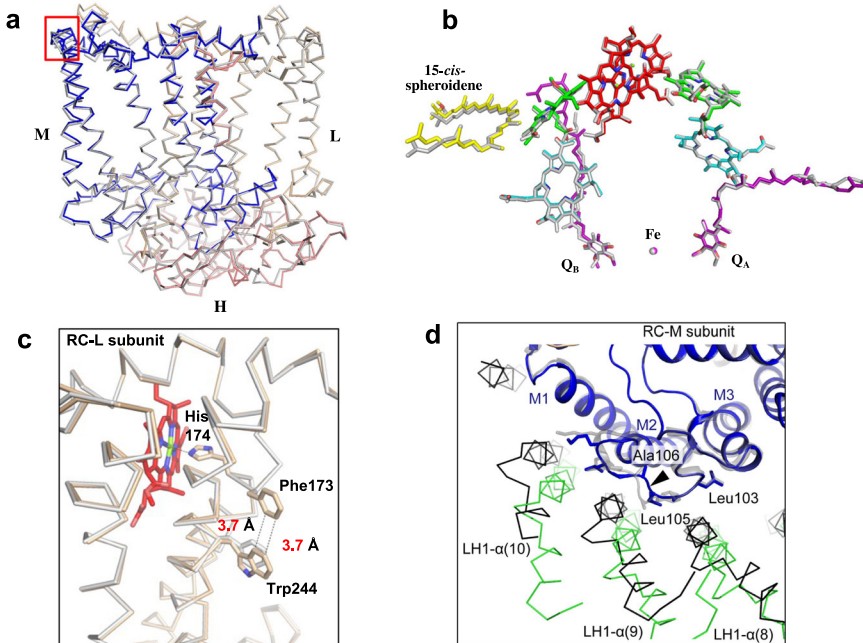

**Fig. 4 | The RC complex of *Rba. capsulatus*. a** Side view of superposition of Cα carbons for the RC proteins between *Rba. capsulatus* (colored) and *Rba. sphaer-oides* (gray, PDB: 7F0L). **b** Cofactor arrangement. Color code: special pair BChl *a*, red; accessory BChl *a*, green; BPhe *a*, cyan; 15-*cis*-spheroidene, yellow; UQ-10, purple; Fe, magenta. **c** Structural comparison of the RC-L subunits for the region around the special pair BChl *a*. Superposition of the Cα carbons of the RC-L sub-units between *Rba. capsulatus* (wheat) and *Rba. sphaeroides* (gray) shows that the Phe173 (next to the special pair BChl *a*-coordinating His174) is in close proximity to Trp244. **d** Structural comparison of the RC-M subunits for the region around the deletion site after conserved Leu105 (arrow head). Superposition of Cα carbons in the RC-M subunits between *Rba. capsulatus* (blue) and the *Rba. sphaeroides* ΔU mutant strain (gray, PDB: 7VY3) shows the different arrangement of LH1-α (*Rba. capsulatus*, green; *Rba. sphaeroides*, black) around the loop between M1 and M2. Viewed from the periplasmic side.

The differences in the special pair BChls *a* between *Rba. capsu-latus* and *Rba. sphaeroides* likely contribute to their different absorp-tion maxima. We have measured light-induced P⁺/P absorption difference spectra of the LH1–RCs purified from *Rba. capsulatus* and *Rba. sphaeroides* strain DP2[25] (Supplementary Fig. 1c). The absorption maxima of the reduced special pair are at 853 nm for *Rba. capsulatus* and 871 nm for *Rba. sphaeroides* in their LH1-assciated forms, indicating that the spectral gap between LH1 and P in *Rba. capsulatus* ($385\,cm^{-1}$) is much larger than that of *Rba. sphaeroides* ($39\,cm^{-1}$). The absorption maxima of the special pairs measured in LH1–RCs are

comparable to those measured using the RC-only complexes (~850 nm for *Rba. capsulatus*[26] and ~865 nm for *Rba. sphaeroides*[27,28]).

The subtle differences in the special pair and surrounding protein environment between *Rba. capsulatus* and *Rba. sphaeroides* may also contribute to their different electron transfer rates within the RC. The *Rba. capsulatus* RC has a faster rate of electron transfer along the B-branch and a slower rate along the A-branch compared with those of *Rba. sphaeroides*[29]. In this connection, sequence alignment of the RC-M subunits of the two species revealed an insertion (Ala) in the loop between helices M1 and M2 in the *Rba. sphaeroides* RC-M (Supplementary Fig. 9). For this reason, the arrangements of three *Rba. capsulatus* LH1 α-polypeptides (8th–10th) around this loop deviate from those in *Rba. sphaeroides* LH1 (Fig. 4d). Conformational changes in the loop caused by this sequence difference likely induced a different packing arrangement of LH1 and RC in the two species' core complexes that would be expected to affect electron transfer rates.

## Discussion

Due to their close phylogenetic relationship and similar spectroscopic properties, *Rba. capsulatus* photocomplexes have long been used as alternatives to those of *Rba. sphaeroides* with the assumption that their corresponding complexes were structurally similar. Although this is generally true for the structurally more conserved RCs, our work has demonstrated that structural differences in the RC-surrounding LH1 complex between these two closely related species are significant and that a broad diversity is possible in the structure of this important component of the photosynthetic apparatus. The crescent-shaped *Rba. capsulatus* LH1–RC containing 10 αβ-subunits represents a major portion of this complex, and this subunit number is comparable with those of a protein-U-lacking *Rba. sphaeroides* mutant (strain ΔU)[15] that contains 10–13 αβ-subunits[8,9]. The ΔU mutant strain has been demonstrated to grow photosynthetically at a similar rate to that of the wild-type[15].

These findings are supported by kinetic analyses that show that the energy transfer rate from LH1 to the reduced RC in *Rba. capsulatus* (~2.8 × 10^10 s^−1)[30] is similar to rates from *Rba. sphaeroides* (~2.7 × 10^10 s^−1, 14 αβ-subunits)[31,32], *Rhodospirillum (Rsp.) rubrum* (1.3–1.7 × 10^10 s^−1, 16 αβ-subunits)[31,33] and *Thermochromatium (Tch.) tepidum* (1.3–2.0 × 10^10 s^−1, 16 αβ-subunits)[34–37]. Thus, these data indicate that the energy transfer rate for LH1 → RC (the rate-limiting step in energy trapping from antennae to the RC) is independent of the number of LH1 αβ-subunits ($N$). Similarly, the *Rba. capsulatus* LH1 structure confirms that the LH1-$Q_y$ absorption maximum is also not significantly dependent on the value of $N$ (*Rba. capsulatus* $Q_y$ = 882 nm versus *Rba. sphaeroides* $Q_y$ = 874 nm and *Rsp. rubrum* $Q_y$ = 877 nm). This agrees with the results of both biochemical analyses and numerical simulation that show that the LH1-$Q_y$ for $N \geq 8$ were all redshifted to a certain wavelength (around 880 nm at room temperature) with respect to that of monomeric BChl *a* (~770 nm)[38–40].

Despite extensive efforts, no dimeric LH1–RC has been isolated from *Rba. capsulatus* (strain Kb-1)[10] or from *Rba. veldkampii*[12,14], and the same was true in our study using the type strain of *Rba. capsulatus* (Supplementary Fig. 1c inset). Among multiple factors, PufX has been demonstrated to play a crucial role in dimerization of the *Rba. sphaeroides* LH1–RC[4–6,41], and its N-terminus has been determined to be directly responsible for the interactions between two monomeric components[6,8,9,22]. The structure in our work here highlights the importance of the PufX N-terminus by showing that the *Rba. capsulatus* PufX N-terminus adopts a remarkably different conformation from those in other *Rhodobacter* species with known structures (Fig. 3), revealing a significant conformational diversity in this region of the protein. This can be understood by the low similarities in overall sequences of PufX (~25% identities for most species)[15] (Supplementary Fig. 8a) and even lower similarities for their N-termini. Indeed, many of the

amino acid residues involved in self-association of *Rba. capsulatus* PufX (Fig. 3d) are not conserved[15]. The self-associated conformation of the N-terminus of *Rba. capsulatus* PufX prevents it from interacting with proteins in another complex, which may explain in part the inability of this species to form a dimeric LH1–RC. Furthermore, the characteristic crescent shape of the 10 αβ-subunit *Rba. capsulatus* LH1–RC may also inherently weaken its ability to form dimers. This conclusion is supported by our previous work showing that the structure of a monomeric LH1–RC from the *Rba. sphaeroides* ΔU mutant strain[15] is similar to that of the native LH1–RC from *Rba. capsulatus*[8] (Supplementary Fig. 10) and that the ΔU mutant strain produced a much smaller amount of dimeric LH1–RC than did wild-type cells[15].

Although protein-U in *Rba. sphaeroides* is unessential for dimerization of LH1–RC, recent studies have shown that protein-U plays an important role in controlling the number of αβ-subunits in an LH1–RC monomer[8,9,15], a prerequisite for both LH1–RC dimer formation and stabilization. *Rba. capsulatus* lacks protein-U, and this likely also contributes to the unique crescent shape of its LH1 and absence of dimeric complexes. The native *Rba. capsulatus* LH1–RC therefore resembles the monomeric LH1–RC from *Rba. sphaeroides* mutant strain ΔU in overall structures[8,9] (Supplementary Fig. 10), with the important distinction that the four αβ-subunits absent from the *Rba. capsulatus* LH1 are present in *Rba. sphaeroides* and associate closely with protein-U in its native *Rba. sphaeroides* LH1–RC (Fig. 1d). Thus, our *Rba. capsulatus* LH1–RC structure provides further evidence that interaction of protein-U with up to four surrounding αβ-subunits is a major mechanism for regulating the number of subunits in the *Rba. sphaeroides* LH1 complexes. This reflects a long-range interaction over a distance of ~ 23 Å (between protein-U and the 11th α-polypeptide) in *Rba. sphaeroides*.

In addition to the effect of protein-U, characteristic interactions between the *Rba. capsulatus* RC-M subunit and nearby LH1 α-polypeptides may also contribute to the unique crescent shape of its LH1. To better elucidate interaction patterns between the M subunit and nearby LH1 α-polypeptides (8th–10th), we mapped interactions of each LH1 α-polypeptide with different colors onto the surface of the M-subunit (Supplementary Fig. 11). In particular, the loop between membrane helices M1 and M2 in the RC-M subunit extensively affects contacts between the M subunit and LH1 α-polypeptides (Supplementary Fig. 11). As a result, the reduced arc size of the *Rba. capsulatus* LH1 weakens its interactions with the RC. This likely explains our ease in isolating an LH1-only complex from photosynthetic membranes of this species during purification (Supplementary Fig. 1) and the results of cryo-EM 3-D classification that showed approximately 5% of all particles to consist of LH1-only complexes (Supplementary Fig. 3a).

Although the smallest native LH1 complex yet discovered, the crescent-shaped *Rba. capsulatus* LH1 structure containing 10 αβ-protein subunits and 37 pigments (20 BChl *a* and 17 spheroidenes) will still provide a challenging target for the rapidly developing tools of protein structure prediction based on artificial intelligence, as represented by AlphaFold2 (AF2)[42]. This algorithm has offered up many promising results with impressive accuracy for the monomeric and oligomeric protein structures composed of a single polypeptide chain[43–45]. Recent AF-Multimer and other packages derived from AF2 have succeeded in predicting structures of several multimeric protein complexes but are still only applicable to the homo-oligomer or hetero-oligomer consisting of a few protein components[46,47]. In order to increase the probability of successfully predicting the structures of hetero-oligomer-containing cofactors, implementation of multimer templates with cofactors and self-distillation approaches is unavoidable, and for such purposes, LH1 structures with variable ring sizes and optimal number of pigments from *Rhodobacter* species are perfect model systems. When the next generation of prediction tools

emerges, it might be possible to deduce the minimal structural requirements for an LH1 complex based on its amino acid sequences and pigment types. Such capability would greatly accelerate progress in dissecting the complex details of the light reactions of photosynthesis−redox biochemistry that ultimately supports all life on Earth.

## Methods

### Preparation and characterization of the LH1−RC complex

A culture of *Rba. capsulatus* NBRC16435 (type strain of the species, ATCC 11166 = DSM 1710 = NCIB 8254) was purchased from National Institute of Technology and Evaluation, Japan. Cells were cultivated phototrophically (anoxic/light) at room temperature for 5 days under incandescent light (60 W). Chromatophores (ICM) at $OD_{850-nm}$ = 40 were treated with 0.25% w/v lauryldimethylamine *N*-oxide in 20 mM Tris-HCl (pH 7.5) at room temperature for 60 min to remove excess LH2, followed by ultracentrifugation at 150,000×*g* at 4 °C for 90 min. LH1−RC complexes were solibilized from the pellets by 1.0% *n*-dodecyl β-D-maltopyranoside (DDM) at room temperature for 60 min in the same buffer, followed by ultracentrifugation at 150,000 × g at 4 °C for 90 min. The crude LH1−RC complexes were loaded onto a DEAE column (Toyopearl 650S, TOSOH) equilibrated at 8 °C with 20 mM Tris-HCl buffer (pH 7.5) containing 0.1% w/v DDM. The fractions were eluted in an order of residual LH2, LH1-only and LH1−RC by a linear gradient of $CaCl_2$ from 0 mM to 100 mM. The LH1−RC and LH1-only fractions were collected and concentrated for absorption and circular dichroism (CD) measurements (Supplementary Fig. 1). LH1−RC complexes were assessed by negative-stain EM using a JEM-1011 instrument (JEOL) (Supplementary Fig. 2a). CD spectra were recorded on a Jasco J-720w spectropolarimeter in a range of 400−1000 nm under the conditions of 100 nm/min scan speed, 5 nm bandwidth, 1 sec response time and 5 repeated scans[48]. Light-induced $P^+/P$ absorption difference spectra of the LH1−RCs were obtained using a V-780 Spectrophotometer (JASCO) by subtracting spectra in the dark from those measured during 940 nm LED illumination at room temperature. Masses of the LH1 polypeptides were determined by matrix-assisted laser desorption/ionization time-of-flight mass spectroscopy (MALDI-TOF/MS)[49] on a 4800 Plus MALDI TOF/TOF Analyzer (Applied Biosystems, MDS SCIEX) equipped with a nitrogen laser (337 nm). Sinapinic acid (3, 5-dimethoxy-4-hydroxy cinnamic acid) was used as matrix and was dissolved in 80% acetonitrile solution containing 0.3% TFA. Typically, the amount of LH1−RC analyzed was of 10−30 pmol. Purified LH1−RC was dissolved in 80% acetonitrile solution containing 0.1% TFA and mixed with the sinapinic acid solution in a ratio of 1:1(v/v) and then loaded onto sample stage for co-crystallization. Analysis was performed in positive and linear modes with typical laser intensity of 4150 and total 1500 shots per spectrum using 4000 Series Explore Software Version 3.5. Phospholipids in intracytoplasmic membranes were analyzed (Supplementary Fig. 6) by $^{31}$P-NMR[50]. The extracted phospholipids were dissolved in 500 µL of chloroform-*d* and mixed with 200 µL of methanol and 50 µL of 0.2 M K-EDTA solution (pH 6.0). $^{31}$P-NMR spectra were recorded at room temperature on a Bruker Biospin Avance III 500 MHz spectrometer equipped with a 5-mm broadband probe tuned to the $^{31}$P nucleus frequency at 202.46 MHz using TopSpin software (ver. 3.6.2). The $^{31}$P-NMR spectra were acquired using inverse-gated proton decoupling (500.13 MHz) with the following parameters: 12,175 Hz sweep width, 30° pulse, 8 K data points, 0.34-s acquisition time, 1.0-s delay time and 2000 scans. $^{31}$P chemical shifts were referred to the peak of 85% $H_3PO_4$. Calibration of the $^{31}$P-NMR signals was carried out using a standard phospholipids kit (Doosan Serdary Research Laboratories, Canada) containing 10 phospholipids.

### Cryo-EM data collection

Proteins for cryo-EM were concentrated to ~3.8 mg/ml. Two and half microliters of the protein solution were applied on a glow-discharged holey carbon grid (200 mesh Quantifoil R2/2 molybdenum), which had been treated with $H_2$ and $O_2$ mixtures in a Solarus plasma cleaner (Gatan, Pleasanton, USA) for 30 s and then blotted and plunged into liquid ethane at −182 °C using an EM GP2 plunger (Leica, Microsystems, Vienna, Austria). The applied parameters were a blotting time of 6 s at 80% humidity and 4 °C. Data were collected on a Titan Krios (Thermo Fisher Scientific, Hillsboro, USA) electron microscope at 300 kV equipped with a Falcon 3 camera (Thermo Fisher Scientific) (Supplementary Fig. 2b). Movies were recorded using EPU software (Thermo Fisher Scientific) at a nominal magnification of 96 k in counting mode and a pixel size of 0.820 Å at the specimen level with a dose rate of 0.846 e-per physical pixel per second, corresponding to 1.26 e-per $Å^2$ per second at the specimen level. The exposure time was 31.8 s, resulting in an accumulated dose of 40 e-per $Å^2$. Each movie includes 40 fractioned frames.

### Image processing

All of the stacked frames were subjected to motion correction with MotionCor2[51]. Defocus was estimated using CTFFIND4[52]. All of the picked particles using the crYOLO[53] were further analyzed with RELION 3.1[54], and selected by 2-D classification (Supplementary Fig. 2c, and Supplementary Table 1). An initial 3-D model was generated in RELION, and the particles were divided into four classes by 3-D classification resulting in only one good class. The 3-D auto refinement without any imposed symmetry (C1) produced a map at 2.67 Å resolution, after contrast transfer function refinement, Bayesian polishing, masking, and post-processing. Then particle projections were further subjected to subtraction of the detergent micelle density followed by 3-D auto refinement to yield the final map with a resolution of 2.62 Å according to the gold-standard Fourier shell correlation (FSC) using a criterion of 0.143 (Supplementary Fig. 3)[55]. The local resolution maps were calculated on RESMAP[56]. Focused 3-D classification[19,20] was conducted for the gap region of the LH ring as well as three LH1 subunits at the open terminal end in an LH1−RC monomer using the mask[57] in RELION until convergence.

### Model building and refinement of the LH1−RC complex

The atomic model of the *Rba. sphaeroides* LH1−RC (PDB: 7F0L) was fitted to the cryo-EM map obtained for the *Rba. capsulatus* LH1−RC using Chimera[58]. Amino acid substitutions and real space refinement for the peptides and cofactors were performed using COOT[59]. Whole regions of PufX and protein−U as well as both terminal regions of the LH1 α-subunit were modeled ab initio based on their density. The manually modified model was refined in real-space on PHENIX[60], and the COOT/PHENIX refinement was iterated until the refinements converged. Finally, the statistics calculated using MolProbity[61] were checked. Figures were drawn with the Pymol Molecular Graphic System (Schrödinger)[62], UCSF Chimera[58], and ChimeraX[63].

### Reporting summary

Further information on research design is available in the Nature Portfolio Reporting Summary linked to this article.

## Data availability

The data that support this study are available from the corresponding authors upon reasonable request. The cryo-EM map has been deposited in the Electron Microscopy Data Bank (EMDB) under accession code EMD-33931 (LH1-RC Complex). Coordinates have been deposited in the Protein Data Bank (PDB) under accession code 7YML (LH1-RC Complex). Previously published structures cited in the work

can be accessed using PDB accession codes 7VY2, 7VY3, 7DDQ, and 7F0L.

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

## Acknowledgements

This research was partially supported by Platform Project for Supporting Drug Discovery and Life Science Research (Basis for Supporting Innovative Drug Discovery and Life Science Research (BINDS)) from AMED under Grant Numbers JP21am0101118 (support number 1758) and JP21am0101116 (support number 1878), and JP22ama121004. R.K., M.H., and B.M.H. acknowledge the generous support of the Okinawa Institute of Science and Technology (OIST), Scientific Computing & Data Analysis Section at OIST and the Japanese Cabinet Office. R.K. acknowledges the support from Prof. Tsumoru Shintake. L.-J.Y. acknowledges support of the National Key R&D Program of China (Nos. 2021YFA0909600 and 2019YFA0904600). This work was supported in part by JSPS KAKENHI Grant Numbers JP20H05086, JP20H02856, and JP22K06111, Takeda Science Foundation, and the Kurata Memorial Hitachi Science and Technology Foundation, Japan.

## Author contributions

Z.-Y.W.-O. and K.T designed the work. K.T., R.K., X.-C. J. I.S., Y.Ko., and M.H. performed the experiments. K.T., R.K., L.-J.Y., Y.Ki., M.T.M., A.M., B.M.H., and Z.-Y.W.-O. analyzed data. Z.-Y.W.-O., K.T., and M.T.M. wrote the paper.

## Competing interests

The authors declare no competing interests.
