## [Peer Review File · Nature Communications]

Rhodobacter capsulatus Forms an Unusually Compact Crescent-Shaped LH1–RC PhotocomplexReviewers' Comments:

Reviewer #1:

Remarks to the Author:

Here, the authors report a high-resolution structure of the LH1-RC complex from the purple bacterium *Rba. capsulatus*. The structure shows several new aspects: (1) the LH1 ring is only 10 subunits instead of the more typical ~18; (2) protein-U, thought to regulate the number of subunits, is also absent; (3) the complex is monomeric and PufX exhibits a distinct conformation of the N-terminus, which has been implicated in dimerization. Overall, these results are important to our understanding of the diverse architectures in purple bacterial photosynthesis and the results are clearly presented. Therefore, I recommend publication and have only a few points that need to be addressed first.

1. The difference in LH1:RC stoichiometry should also be visible in the UV spectrum. Can the authors compare the relative peak heats for the complex from *Rba. capsulatus* and *Rb. sphaeroides* to show the difference in stoichiometry is present in the full sample? This would confirm the decrease in subunit stoichiometry isn't related to a disordered organization of part of the ring or some other structural consideration.
2. The authors state that the extra BChl observed in the structure is not involved in energy transfer because it is >2 nm from other BChl. However, chromophores at that distance or even further apart can certainly participate in energy transfer (e.g., the entire field of FRET as a spectroscopic ruler for distances of 1-10 nm). The authors should correct this statement and clarify what the nearest BChl are that could be energetically connected.
3. The authors state that LH1 to RC energy transfer occurs at the same rate even with fewer subunits, as expected. In contrast, the decrease in number of subunits is expected to decrease the rate of RC to LH1 energy transfer because of fewer acceptors. This type of back transfer is hypothesized to allow the excitation to reach other RCs in the case where the RC is closed. Related to this point,
 - a. Can the authors say something about other strategies to allow the excitation to escape?
 - b. Does this organism grow in an ecological niche where the light is low enough that the presence of multiple excitations is very, very rare?
 - c. Could the newly discovered cofactor play a role in mediating back transfer?
4. The authors describe how the N-helix of the Puf-X protein is positioned so that it doesn't mediate dimer formation. Can they clarify what is lacking as compared to the case where the N-helix does serve to mediate dimer formation?

Reviewer #2:

Remarks to the Author:

The manuscript by Tani et al. presents a cryo-EM structure of the monomeric LH1-RC complex from *Rba. capsulatus*. Unlike other similar complexes that typically have 14-15 LH1s, the *Rba. Capsulatus* complex contains only 10 LH1 subunits distributed on one side of the RC, forming a crescent-shaped complex. The lack of terminal LH1 subunits is attributed to the absence of protein U and a single deletion in the loop region of the RC-M subunit in *Rba. capsulatus*. However, previous studies by the same group and others have demonstrated that protein U is crucial for the association of terminal LH1s with RC (Tani, K. et al. *Nat. Commun.* 2022; Cao, P. et al. *Nature Commun.* 2022). Complexes of RC encircled by incomplete LH1 ring were observed in protein U-deleted mutant strains of *Rba. Sphaeroides*, and their structures were reported. In fact, Fig. S9 clearly shows that the overall structure of *Rba. Capsulatus* LH1-RC is very similar to that of LH1-RC purified from the protein U-deleted *Rba. Sphaeroides* strain (PDB: 7VY3). Therefore, the *Rba. Capsulatus* seems to represent the mutant version of protein U-deleted *Rba. Sphaeroides*, as does its LH1-RC complex. In addition, the authors found that compared with other PufX-containing species, PufX in *Rba. capsulatus* exhibits a different conformation at its N-terminus, which prevents the formation of dimeric LH1-RC complex. This is not new, as previous studies have well established that PufX, especially its N-terminus, plays a crucial role in mediating LH1-RC dimerization in *Rhodobacter* Species.

Therefore, although this work reports some interesting features of the *Rba. capsulatus* LH1-RC complex, this reviewer thinks that the novelty of this work is low.

Other comments

Please clarify whether the *Rba. Capsulatus* LH1-RC complex contains exactly 10 pairs of LH1, or extra LH1 subunits are present but not identified/recognized because of weak binding and/or poor density.

Response to reviewers:

Reviewer #1

Reviewer #1's comments: Point 1

Here, the authors report a high-resolution structure of the LH1-RC complex from the purple bacterium *Rba. capsulatus*. The structure shows several new aspects: (1) the LH1 ring is only 10 subunits instead of the more typical ~18; (2) protein-U, thought to regulate the number of subunits, is also absent; (3) the complex is monomeric and PufX exhibits a distinct conformation of the N-terminus, which has been implicated in dimerization. Overall, these results are important to our understanding of the diverse architectures in purple bacterial photosynthesis and the results are clearly presented. Therefore, I recommend publication and have only a few points that need to be addressed first.

1. The difference in LH1:RC stoichiometry should also be visible in the UV spectrum. Can the authors compare the relative peak heights for the complex from *Rba. capsulatus* and *Rb. sphaeroides* to show the difference in stoichiometry is present in the full sample? This would confirm the decrease in subunit stoichiometry isn't related to a disordered organization of part of the ring or some other structural consideration.

Our response:

We appreciate the reviewer's positive assessment of our work. A comparison of the UV spectra normalized at LH1- Q_y for the LH1-RCs purified by DEAE chromatography between *Rba. capsulatus* and *Rba. sphaeroides* is shown in Fig. R1 below as requested by the reviewer. In this case, the absorbance at ~800 nm from the RC accessory BChl in the *Rba. capsulatus* LH1-RC is higher than that of *Rba. sphaeroides* LH1-RC. However, we should point out that although such a comparison has been often used for a rough estimate, it cannot be used as an accurate method for calculating the stoichiometry of LH1-subunit:RC due to (i) the multi-component nature of the absorption spectra including overlap between the RC and LH1 BChls and overlap between LH1- Q_y and trace-amount LH2 peak, and (ii) the large difference in the absorption intensities between LH1- Q_y and the 800-nm peak. As a result, small changes in the 800-nm peak can result in large differences in the LH1-subunit:RC stoichiometry because the 800-nm intensity is strongly affected by the purity of LH1-RC and the closely spaced LH1- Q_y peak. The inability of such measure has been shown in the case of protein-U-deleted *Rba. sphaeroides* LH1-RC (Fig. R2, cited from Ref. 15 Supplementary Fig. 8b), in which essentially identical absorption spectra were observed for both the highly purified ΔU -monomeric (11 α and 10 β per RC) and wild-type (14 $\alpha\beta$ per RC) LH1-RCs.

Fig. R1 Absorption spectra of the purified LH1-RCs from *Rba. capsulatus* and *Rba. sphaeroides*.

Fig. R2 Absorption spectra of the ΔU and WT LH1-RCs purified from *Rba. sphaeroides* IL106. Cited from Ref. 15 Supplementary Fig. 8b.

Reviewer #1's comments: Point 2

2. The authors state that the extra BChl observed in the structure is not involved in energy transfer because it is >2 nm from other BChl. However, chromophores at that distance or even further apart can certainly participate in energy transfer (e.g., the entire field of FRET as a spectroscopic ruler for distances of 1-10 nm). The authors should correct this statement and clarify what the nearest BChl are that could be energetically connected.

Our response:

The extra BChl in the *Rba. capsulatus* LH1-RC was first identified in this work. Because its function is unclear, our statement was based solely on speculation. However, according to the reviewer's suggestion, we have modified this statement and added descriptions on the nearby BChls that could serve as potential partners in energy transfer.

Reviewer #1's comments: Point 3

3. The authors state that LH1 to RC energy transfer occurs at the same rate even with fewer subunits, as expected. In contrast, the decrease in number of subunits is expected to decrease the rate of RC to LH1 energy transfer because of fewer acceptors. This type of back transfer is hypothesized to allow the excitation to reach other RCs in the case where the RC is closed. Related to this point,

- a. Can the authors say something about other strategies to allow the excitation to escape?
- b. Does this organism grow in an ecological niche where the light is low enough that the presence of multiple excitations is very, very rare?
- c. Could the newly discovered cofactor play a role in mediating back transfer?

Our response:

- It is our belief that the spectral overlap between LH1 and RC may be more important for excitation energy transfer than the absolute number of LH1 subunits. Although *Rba. capsulatus* LH1 has a fewer number of subunits, it exhibits an LH1- Q_y at 882 nm similar to those of typical LH1 complexes, indicating a strong coupling between the LH1 BChls. This is supported by the shorter Mg-Mg distances measured between the *Rba. capsulatus* LH1 BChls

than between those in *Rba. sphaeroides* monomeric LH1 (Supplementary Table 2). The strong coupling in the *Rba. capsulatus* LH1 BChls is thought to facilitate excitation energy transfers both forward and backward between LH1 and the RC at least as efficient as those of other LH1s. It has been demonstrated in Ref. 30 that most *Rba. capsulatus* RCs that are directly excited undergo charge separation and not backward energy transfer to the LH1 antenna complexes. However, to further address the reviewer's concern, we have measured light-induced P⁺/P absorption difference spectra of the purified LH1–RCs from *Rba. capsulatus* and *Rba. sphaeroides* and have added it to the manuscript as Supplementary Fig. 1c. These data show that absorption of the reduced special pair (P) is at 853 nm for *Rba. capsulatus* and 871 nm for *Rba. sphaeroides* both in their LH1-associated forms, indicating that the spectral gap between LH1 and P (882–853 = 29 nm) in *Rba. capsulatus* is much larger than that (874–871 = 3 nm) of *Rba. sphaeroides*. This implies a much more favorable “downhill” energy transfer from the RC to LH1 in *Rba. capsulatus*, and may represent an adaptive strategy for balancing the forward and backward energy transfers between LH1 and RC. Such a regulating mechanism is ultimately reflected in the ability of this organism to grow phototrophically. A similar case has been observed for the protein-U-deleted *Rba. sphaeroides* mutant that grows phototrophically at a similar rate to that of the wild-type despite having only 11 α - and 10 β -polypeptides in its LH1 (Ref. 8, 15).

- *Rba. capsulatus* was isolated from a freshwater habitat and grows in aquatic environments similar to those of other typical purple phototrophs. Many of these environments are light-limiting for oxygenic phototrophs, but anoxygenic phototrophs, with their capacity to harvest light at very low intensities, flourish there. As mentioned above, a fewer number of LH1 subunits would not compromise the kinetics of excitation energy transfer between LH1 and RC through adjusting their spectral behavior over varied light conditions.
- The extra BChl newly identified in the *Rba. capsulatus* LH1–RC might play a role in excitation energy transfer due to its unique position between LH1 and RC as pointed out by the reviewer. However, it could be a challenging task to experimentally verify the function(s) of this extra BChl because of the difficulties in extracting its spectroscopic component from the heavily overlapped absorption region occupied by the large number of BChls in LH1 and RC. Assuming that an unambiguous result is actually possible in this regard, a highly selective detection method that specifically targeted this extra BChl would need to be devised to study this problem. To our knowledge, such a method has not been reported.

Reviewer #1's comments: Point 4

4. The authors describe how the N-helix of the Puf-X protein is positioned so that it doesn't mediate dimer formation. Can they clarify what is lacking as compared to the case where the N-helix does serve to mediate dimer formation?

Our response:

As we mentioned in the Discussion section, identities of the amino acid sequence for PufX are low (~25% for most species) (Ref. 15) and even lower for their N-termini. At present, dimeric LH1–RC structures have been determined from only one species (*Rba. sphaeroides*). Therefore, it is difficult at the current stage to identify the specific residues (or fragment, motif) in the PufX N-terminus that are responsible for dimerization, if such a fragment or motif actually exists. In addition, although PufX is crucial for dimerization of LH1–RC, we have demonstrated that protein-U also contributes to dimer formation, deleting protein-U from *Rba. sphaeroides* resulted in a significant decrease in dimer proportion. To clarify the minimal requirements for dimer formation, more structures of both dimeric and monomeric LH1–RCs are required.

Reviewer #2

Reviewer #2's comments

The manuscript by Tani et al. presents a cryo-EM structure of the monomeric LH1-RC complex from *Rba. capsulatus*. Unlike other similar complexes that typically have 14-15 LH1s, the *Rba. Capsulatus* complex contains only 10 LH1 subunits distributed on one side of the RC, forming a crescent-shaped complex. The lack of terminal LH1 subunits is attributed to the absence of protein U and a single deletion in the loop region of the RC-M subunit in *Rba. capsulatus*. However, previous studies by the same group and others have demonstrated that protein U is crucial for the association of terminal LH1s with RC (Tani, K. et al. Nat. Commun. 2022; Cao, P. et al. Nature Commun. 2022). Complexes of RC encircled by incomplete LH1 ring were observed in protein U-deleted mutant strains of *Rba. Sphaeroides*, and their structures were reported. In fact, Fig. S9 clearly shows that the overall structure of *Rba. Capsulatus* LH1-RC is very similar to that of LH1-RC purified from the protein U-deleted *Rba. Sphaeroides* strain (PDB: 7VY3). Therefore, the *Rba. Capsulatus* seems to represent the mutant version of protein U-deleted *Rba. Sphaeroides*, as does its LH1-RC complex.

In addition, the authors found that compared with other PufX-containing species, PufX in *Rba. capsulatus* exhibits a different conformation at its N-terminus, which prevents the formation of dimeric LH1-RC complex. This is not new, as previous studies have well established that PufX, especially its N-terminus, plays a crucial role in mediating LH1-RC dimerization in *Rhodobacter* Species.

Therefore, although this work reports some interesting features of the *Rba. capsulatus* LH1-RC complex, this reviewer thinks that the novelty of this work is low.

Other comments

Please clarify whether the *Rba. Capsulatus* LH1-RC complex contains exactly 10 pairs of LH1, or extra LH1 subunits are present but not identified/recognized because of weak binding and/or poor density.

Our response:

- Although overall structures of the LH1-RCs from wild-type *Rba. capsulatus* and the ΔU strain of *Rba. sphaeroides* look similar, the structural details between the two complexes are quite different. As we mentioned in the last paragraph of the Discussion section, interactions between the *Rba. capsulatus* RC-M subunit and nearby LH1 α -polypeptides are unique and likely contribute to the characteristic crescent shape of its LH1 (Supplementary Fig. 11). In particular, the loop between membrane helices M1 and M2 in the RC-M subunit extensively affects contacts between the M subunit and LH1 α -polypeptides. As a result, the reduced arc size of the *Rba. capsulatus* LH1 weakens its interactions with the RC. These structural features differ from those in the LH1-RCs from *Rba. sphaeroides* (both native and ΔU strains) and *Rba. veldkampii* (Supplementary Fig. 11b).
- Although previous biochemical analyses showed that the *Rba. sphaeroides* PufX N-terminus plays an important role in dimer formation, it was unclear why the PufX-containing *Rba. capsulatus* only forms monomeric LH1-RC despite the close phylogenetic relationship and similar spectroscopic behavior of the two species. Our structural analysis on the *Rba. capsulatus* LH1-RC has revealed a structural basis for this, and, more importantly, provides an example of the danger in making broad-sweeping generalizations about features of even closely related species without first doing the experiment (*Rba. capsulatus* photocomplexes have long been used as alternatives to those of *Rba. sphaeroides* under the assumption that their corresponding complexes are structurally similar).
- To address the reviewer's concern, we have calculated 3D classification in a localized region (containing the LH1 gap and three $\alpha\beta$ -subunits at one end of the LH1 crescent) known as focused classification (Ref. 19, 20) using the cryo-EM program suite RELION, which could reveal small distinctions between 3D classes. The results have been added as Supplementary Fig. 4. Four classes of the 3D maps were computed until convergence, all of them showing similar conformations independent of their resolutions. After checking these maps carefully, only a single class in each form showed well-resolved density corresponding to the three LH1

subunits. By contrast, other maps were relatively featureless but were unable to accommodate additional LH1 subunits. We have added these descriptions in the revised text (Results and Methods). However, due to the inherent heterogeneities in the form of LH1 complexes as revealed by 2D and 3D classifications (Supplementary Figs. 3 and 4), we cannot exclude the possibility that other LH1–RC forms could be obtained using different purification methods or may exist in other purple bacterial species. Thus, to be cautious, we have modified the title and relevant expressions in the revised manuscript to remove our previous speculation that the LH1–RC photocomplex from *Rba. capsulatus* is a “minimalist” form of this structure.

Reviewers' Comments:

Reviewer #1:

Remarks to the Author:

The authors have addressed many of my comments, adding interesting information about the implications and biological relevance of the different structures.

However, they did not expand this discussion in the manuscript itself. I would encourage the authors to do so, otherwise the importance of the results may not be clear to the community as evidenced by both reviews.

Reviewer #2:

Remarks to the Author:

This manuscript reports structural features of the monomeric LH1-RC complex from *Rba. capsulatus* in great details. The revised manuscript is clearly written and much improved. The authors responded adequately to the comments by this reviewer.

This reviewer agrees that the manuscript reports some unique features of this complex and the work is well done, however, it is also true that this complex is quite similar with that from the *Rba. sphaeroides* deltaU strain. Therefore, this reviewer still thinks that the novelty of this work is low.

Revisions:

- Abstract has been shortened.
- A paragraph has been added at the end of Discussion to emphasize the interesting nature of the results presented to a broader structural biology community, according to the editor's and reviewer #1's suggestions.
- Additional information on the legend of Supplementary Fig. 2a-b has been provided.